# Research Progress of Single-Photon Emitters Based on Two-Dimensional Materials

**DOI:** 10.3390/nano14110918

**Published:** 2024-05-23

**Authors:** Chengzhi Zhang, Zehuizi Gong, Dawei He, Yige Yan, Songze Li, Kun Zhao, Jiarong Wang, Yongsheng Wang, Xiaoxian Zhang

**Affiliations:** Key Laboratory of Luminescence and Optical Information, Ministry of Education, Institute of Optoelectronic Technology, School of Physical Science and Engineering, Beijing Jiaotong University, Beijing 100044, China; 23121752@bjtu.edu.cn (C.Z.); 20271236@bjtu.edu.cn (Z.G.); dwhe@bjtu.edu.cn (D.H.); 22121683@bjtu.edu.cn (Y.Y.); 22121666@bjtu.edu.cn (S.L.); 20118047@bjtu.edu.cn (K.Z.); 22110527@bjtu.edu.cn (J.W.)

**Keywords:** single-photon emitter, 2D materials, TMDs, Purcell enhancement, quantum emitters

## Abstract

From quantum communications to quantum computing, single-photon emitters (SPEs) are essential components of numerous quantum technologies. Two-dimensional (2D) materials have especially been found to be highly attractive for the research into nanoscale light–matter interactions. In particular, localized photonic states at their surfaces have attracted great attention due to their enormous potential applications in quantum optics. Recently, SPEs have been achieved in various 2D materials, while the challenges still remain. This paper reviews the recent research progress on these SPEs based on various 2D materials, such as transition metal dichalcogenides (TMDs), hexagonal boron nitride (hBN), and twisted-angle 2D materials. Additionally, we summarized the strategies to create, position, enhance, and tune the emission wavelength of these emitters by introducing external fields into these 2D system. For example, pronounced enhancement of the SPEs’ properties can be achieved by coupling with external fields, such as the plasmonic field, and by locating in optical microcavities. Finally, this paper also discusses current challenges and offers perspectives that could further stimulate scientific research in this field. These emitters, due to their unique physical properties and integration potential, are highly appealing for applications in quantum information and communication, as well as other physical and technological fields.

## 1. Introduction

Quantum mechanics has unveiled the portal to understanding the microscopic realm, revealing a series of peculiar and counterintuitive phenomena within nature, such as wave–particle duality and entangled states. This theory not only lays the foundation for our comprehension of the essence of matter and the architecture of the cosmos but also exerts a profound influence across numerous scientific and technological domains. For instance, the band theory, predicated on quantum mechanics, has elucidated the existence of semiconductors. To this day, quantum mechanics is regarded as one of the most triumphant theories in physics, and the application of these astonishing quantum effects to realize fascinating quantum technologies has become a tangible reality. By capitalizing on the information encapsulated within intrinsic quantum systems, an array of quantum technologies with vast potential is burgeoning, encompassing quantum informatics, quantum metrology, and quantum imaging, among others [1,2,3,4]. Meanwhile, the exceptional attributes of photons, including the speed of light propagation, low noise, and the ease of manipulating polarization, have allowed quantum light sources to shine in these domains. Concurrently, the generation, detection, and manipulation of quantum light sources and their quantum states have become foundational to these quantum technologies. Within this realm, SPEs, as specialized quantum light sources, are increasingly being integrated into quantum technologies [5,6]. Their most notable characteristic is the ability to emit a photon on demand within a specified time interval. This feature has found widespread application in various quantum technologies, such as in the design of two-qubit entangled logic gates for quantum key distribution (QKD) [7] and as carriers of qubits in quantum communication [8].

A majority of quantum technologies necessitate a high-quality SPE to fully realize their potential. An ideal SPEs should emit a specific photon on demand at a designated time, with the probability of emitting a second photon being zero [9]. Additionally, the emitter is characterized by an exceedingly brief interval between repeated excitations, and the subsequent photons it releases are indistinguishable. These criteria ultimately manifest in three parameters that gauge the performance of a single-photon emitter: purity, brightness, and indistinguishability [10,11,12,13]. The parameter of purity serves to characterize the likelihood that an SPE emits only one photon per instance; thus, a higher purity signifies a greater probability of emitting a single photon, and a lower probability of multiple photon emissions. Typically, the purity of an SPE is quantified by the value of the second-order coherence function at zero delay, g^2^(0), obtained from the Hanbury Brown and Twiss (HBT) experiment. The principle of the HBT experiment involves splitting the light field emitted from an SPE into two non-interfering parts, using a beam splitter. These parts are then transmitted through optical fibers to two photon detectors, which record the arrival times of the photons. The data from these detectors are subsequently sent to a coincidence counter to calculate the second-order coherence function, ultimately producing the g^2^(τ) graph, where τ represents the time difference between the detections by the two photon detectors [14,15]. For an ideal SPE, since it emits only one photon, it is impossible for two photon detectors to simultaneously detect a photon. Therefore, in the resulting g^2^(τ) graph, a pronounced dip at τ = 0 will be observed, where the value of g^2^(0) ideally equals zero. However, due to background noise or insufficient detection precision, g^2^(0) will not be zero. It is generally accepted that if g^2^(0) is less than 0.5, the source can be considered to emit a single photon per cycle. This phenomenon, where g^2^(0) is less than 0.5, is commonly referred to as the antibunching of photons. The second parameter is brightness. In the context of an SPE, due to its inherent fragility, the light pulses it emits may suffer losses in the environment or optical components, leading to the occurrence of vacuum states (no photons in a wave packet). Concurrently, the time interval for repeated excitation of the SPE is generally not zero. Therefore, to characterize how many photons are actually excited within a certain period, brightness can be used as an indicator. Generally, brightness can be simply represented by the light intensity detected by the detector. Lastly, indistinguishability refers to the identical nature of the photons emitted by the SPE, which is typically measured by the indistinguishability, M, obtained from the HOM experiment [13].

Currently, a multitude of systems are utilized to fabricate SPEs, all of which operate on similar principles. These involve a system encompassing a localized ground state and an excited state energy level. The energy of these two levels should ideally be as isolated from the system as feasible. Furthermore, these two levels must possess identical spin to adhere to the spin conservation condition during light excitation and radiation [16]. Initially, researchers coupled individual atoms [17,18], ions [19,20], and cavities or excited individual molecules to generate SPEs [21,22]. However, these types of SPEs have issues such as a limited atomic capture time and competition with cavity modes. Subsequently, solid-state systems such as quantum dots and color centers began to be extensively studied for single-photon emission [23,24,25,26,27,28,29]. Among the best-performing SPEs in solid-state systems are the III-V quantum dots, like InGaAs quantum dots and color center in Diamond. Thanks to the solid state environment, these single-photon emitters have better stability, in which quantum dots can also be coupled to optical cavities for performance enhancement, as in the case of atomic molecule-based single-photon emitters; however, the drawbacks of solid state environments are also obvious, as higher refractive indices make it difficult for photons to penetrate through the bulk material to reach the surface, which results in relatively low coupling efficiency. At the same time, the complex production process also affects the application of single-photon emitters in solid environments to a certain extent. In recent years, two-dimensional (2D) materials have emerged as a hot topic in the field of SPEs [30], such as TMDs and wide bandgap materials primarily based on hBN. For a 2D system, due to its atomic-scale thickness and strong in-plane bonds, the photons it emits can be easily extracted, making it easier to integrate with various optical components, such as waveguides and optical fibers [31,32]. Furthermore, the strong in-plane bonds allow for band modification through local strain engineering [33], a feature not present in other systems. Simultaneously, the degree of twist freedom between 2D materials provides a novel approach for exciton localization, a necessity for SPEs. In summary, the rich characteristics of 2D materials have paved a new path for research in the field of SPEs, making the fabrication, positioning, and performance control of SPEs more promising.

This article summarizes the research advancements in various 2D layered materials for SPEs, initially identifying which are conducive for such applications. It then outlines the methodologies for achieving single-photon emission and positioning of SPEs, followed by an analysis of strategies to enhance their performance. Lastly, it offers a forward-looking perspective on the future development, potential applications, and challenges of SPEs.

## 2. SPEs Based on 2D Materials

When light strikes the surface of a material and its frequency meets the criteria for electronic transitions, the energy is absorbed by electrons, elevating them to higher energy states. This excitation results in the formation of free electron–hole pairs within the material. However, when the Coulomb interaction between the electrons and holes is considered, these free pairs coalesce into a composite entity known as an exciton. Within 2D material systems, the reduced Coulomb screening between electrons and holes results in a significant increase in exciton binding energy [34], enabling the formation of stable exciton structures even at room temperature [35,36]. Consequently, the optical properties of these materials are predominantly governed by excitons. Interestingly, in TMDs, reducing the thickness to a single layer transforms the bandgap from indirect to direct [37], substantially enhancing the efficiency of exciton recombination. Moreover, in TMDs, the spatial symmetry of the single layer is disrupted, leading to inequivalent K and K′ valleys [38,39]. This asymmetry allows only left-handed or right-handed circularly polarized light to excite electrons in these specific valleys, creating excitonic valley degrees of freedom. These characteristics have garnered significant interest in the fields of photonics and valleytronics. Currently, the localization of exciton wavefunctions in TMDs reveals their capability for single-photon emission [40], while the rich excitonic properties of these materials also manifest in diverse and intriguing photophysical properties in SPEs. Additionally, the Moiré superlattice structure in TMDs facilitates the further localization of excitons, enhancing their prominence in the field of quantum optics.

On the other hand, the optical active defect levels within the wide bandgap of hBN also achieve single-photon emission. The wide bandgap not only allows for single-photon emission at room temperature but also provides a considerable range of tunability for the emission wavelength. These characteristics make it a promising candidate for scalable nanophotonic circuits on a single chip [41].

### 2.1. SPEs Based on TMDs

In 2015, localized emitters in monolayer WSe_2_ were first observed at low temperatures [42], appearing randomly along the sample edges (Figure 1a). These emitters exhibited photoluminescence (PL) spectra with extremely narrow full width at half maximum (FWHM) of about 120 μeV. Subsequent HBT experiments confirmed the single-photon emission characteristics of these localized emitters. The main emission peak energies of these emitters were red-shifted by approximately 40–200 meV compared to the neutral bright excitons in WSe_2_ [43,44,45]. Polarization-dependent PL experiments revealed the linear polarization characteristics of these emission peaks [42,46,47]. Additionally, some studies found a significant decrease in the PL intensity of neutral bright excitons at the localized emitters’ positions, suggesting that these localized emitters might originate from neutral excitons trapped by potential wells formed by structural defects or local strains [42]. Interestingly, the quantum emitters in WSe_2_ inherit many of the intricate structural characteristics of neutral excitons. For instance, the split peaks in certain WSe_2_ quantum emitters exhibit features of the neutral exciton intervalley hybridization doublet [46]. The polarization-resolved PL spectrum reveals the linearly polarized emission characteristics of these hybridization doublets [46]. Similarly to neutral excitons, applying an external magnetic field can mitigate the exchange interactions between electrons and holes [45,48], thus restoring valley polarization emission properties and transforming cross-linearly polarized emissions into cross-circularly polarized emissions [46]. Notably, most localized emitters in monolayer WSe_2_ exhibit a pronounced Zeeman effect [43,46], with g-factors around 8–10 [43,44,45,46], ten times that of InAs quantum dots. Due to the electron–phonon exchange interaction under an anisotropic magnetic field, the zero-field splitting energy is approximately 0.71–0.73 meV [43]. Recent theories suggest that these emitters with high g-factors may arise from localized defect excitons formed by the hybridization of dark excitons with defect states [49,50]. Intriguingly, additional resonance peaks, BS-X, were found in the photoluminescence excitation (PLE) spectra of some localized emitters in monolayer WSe_2_ [51], blue-shifted by about 5meV relative to the ground state exciton energy. When the excitation light energy was tuned to resonate with the BS-X peak [51], the g^2^(0) of the localized emitters could be less than 0.002 (Figure 1b) [51]. It is worth mentioning that the bunching phenomenon demonstrated by cross-correlation experiments confirmed the biexciton cascade emission characteristics of the localized emitters at the edges of WSe_2_ flakes [50,52], with the power dependence of the light intensity of the two emission peaks showing clear sublinear and superlinear behaviors [52]. The fine structure splitting (FSS) of the biexciton state and the corresponding cross-polarization phenomena were also observed in polarization-correlated PL spectroscopy experiments [50,52], with an FSS of about 0.4 meV (Figure 1c) [52]. Finally, separate HBT experiments verified the single-photon emission properties of each peak in the cascade emission. Interestingly, due to the strong in-plane atomic bonds and superior mechanical properties, the SPEs in WSe_2_ can be tuned in emission wavelength by applying external strain or pressure [53,54], such as through a 2D piezoelectric platform, achieving reversible tuning of the emission energy of monolayer WSe_2_ up to 18 meV (Figure 1d) [53].

For W-based TMDs (WSe_2_ and WS_2_), the spin orientations of the highest valence band and the lowest conduction band are antiparallel [51], resulting in a dark excitonic ground state that significantly limits the brightness of SPEs. However, with advancing research, similar single-photon emissions have been discovered in M-based TMDs [55,56,57]. For instance, in monolayer MoSe_2_, by utilizing local strain to bind bright neutral excitons or charged excitons, a single quantum emitter with a radiative rate higher than 1 ns^−1^ can be achieved (Figure 2a) [57]. In contrast, the radiative rate of WSe_2_ is merely 0.01 ns^−1^. Simultaneously, single-photon emissions have been observed on monolayer MoS_2_ and MoTe_2_ [58]. Notably, multilayer MoTe_2_ can even generate single-photon emissions within the telecom wavelength [56].

In conclusion, it is noteworthy that, within TMDs, the Moiré superlattice structure emerges when different or identical 2D materials are stacked at slight twist angles. This minute interlayer twist or lattice mismatch induces a new large periodic structure atop heterostructures or homostructures, known as the Moiré pattern. Such periodic patterns lead to periodic variations in the bandgap of the heterostructures or homostructures [59], and the periodic changes in the bandgap can significantly influence the behavior of electrons and excitons within the material, endowing them with an additional periodic potential field. The lowest points of this potential field are located at the positions of high symmetry within the Moiré superlattice structure [60]. Furthermore, the depth of the potential field can be modulated by altering the twist angle, with a maximum variation of several hundred megaelectron volts. Excitons, once trapped at these points of high symmetry, have their energy modulated by the potential field, resulting in localized Moiré excitons. In spectroscopic terms, this is observed as the splitting of the exciton peak into a series of equidistant ultranarrow peaks [59,61], as shown in Figure 2b, positioning it as a potential candidate for SPEs. In MoSe_2_/WSe_2_ Moiré superlattices with a twist angle of 2°, researchers have observed interlayer excitons trapped by the Moiré potential. At a low excitation power of 20 nW, the broad interlayer photoluminescence (PL) spectrum developed several narrow peaks around the free interlayer exciton energy of approximately 1.33 eV, with an average linewidth of 100 μeV [62], similar to the linewidth of the aforementioned SPEs. Subsequent experiments discovered that the interlayer excitons captured in MoSe_2_/WSe_2_ Moiré superlattices with twist angles close to 60° exhibit single-photon emission characteristics (Figure 2c) [63]. Unlike SPEs of monolayer WSe_2_, these photons originate from interlayer excitons, thus possessing a larger permanent out-of-plane dipole moment. This allows for the tuning of exciton emission energies through the Stark effect, as shown in Figure 2d. Moreover, under low excitation densities, a strong Moiré potential can confine individual excitons within Moiré cells, forming an array of quantum emitters.

**Figure 2 nanomaterials-14-00918-f002:**
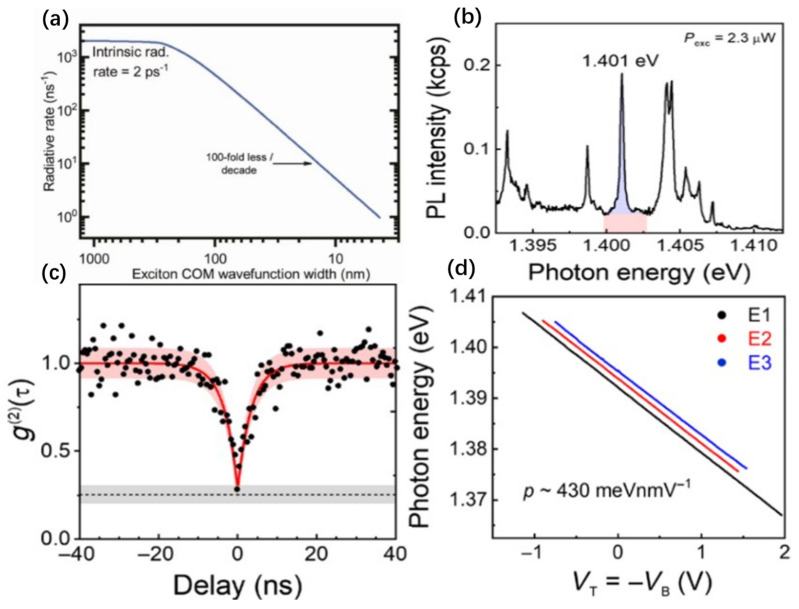
SPEs of other TMDs. (**a**) Radiative rate in the presence of exciton localization using established theories for semiconductor nanostructures. The exciton COM wave function (1/e^2^) width is determined by the form of the harmonic confinement potential. The radiative rate is greater than ~1 ns^−1^ for all plausible confinement scenarios (i.e., width ≥ 4 nm), thus exceeding that of WSe_2_ QEs by 2 orders of magnitude. (**b**) PL spectrum from MoSe_2_/WSe_2_ Moiré superlattices. The blue and red regions represent the estimated PL signal from the emitter and the background. (**c**) Plot of g^2^(τ) measurements of the 1.401 emission peak in Figure 2b, using 2.3 μW CW excitation at 760 nm, show clear antibunching. The red solid line represents a fit of the experimental data, revealing a g^2^(0) = 0.28 ± 0.03. The black dots are the data measured by the HBT experiment and the red curve is the fitted result. The black dashed line represents the experimental limitation for g^2^(0) owing to the nonfiltered emission background. (**d**) Plot of emission energy versus gate voltage for E1, E2, and E3 peaks. Values of electrical dipole moments of ~420 meVnmV^−1^ are determined by the linear fits. Panel (**a**) reprinted with permission from Ref. [57], Copyright 2021, American Chemical Society. Panels (**b**–**d**) reprinted with permission of AAAS from Ref. [63]. Copyright 2020, the authors, some rights reserved, exclusive licensee AAAS. Distributed under a Creative Commons Attribution License 4.0 (CC BY).

### 2.2. SPEs Based on hBN

Wide-bandgap hBN is considered an ideal material for SPEs operable at room temperature, primarily due to the high degree of localization of certain defect energy levels and the ample bandgap that provides sufficient scope for tuning the emitted photon energies. Single-photon emission has been observed in bulk, multilayer, and monolayer hBN [41,64], with multilayer hBN defects exhibiting sharp zero phonon lines (ZPLs) and broader phonon sidebands (PSB) at room temperature (Figure 3a) [41,65,66]. The zero-phonon line energies of these defects are distributed between 1.6 eV and 2.2 eV due to local strain and varying dielectric environments [65,67]. Most hBN defects’ ZPL shapes are asymmetric due to phonon interactions [41,65,66,67], and some defect spectra have a Debye–Waller factor (the intensity ratio of the ZPL to the PSB) of up to 0.82 [41]. The saturation intensity of SPEs at room temperature can reach 4.2 × 10^6^ s^−1^ [41]. These single photons are believed to originate from N_B_V_N_ defects in hBN [41]. Interestingly, polarization-dependent photoluminescence (PL) measurements indicate linear polarization characteristics of hBN defect zero-phonon lines [41,65,68], although the orientations of the excitation and emission dipoles are not always aligned [65,68,69,70], possibly due to cross-relaxation between defects [71]. Additionally, some quantum emitters in hBN exhibit nonlinear excitation phenomena [72], where two photons with wavelengths twice that of the single-photon wavelength are absorbed, making them promising for use in bioimaging applications [72]. While the emission wavelengths of certain SPEs in hBN thin films are unaffected by magnetic fields [69], their luminescence intensity shows a clear magnetic field dependence (Figure 3b) [69], being strongest (weakest) when the field direction is perpendicular (parallel) to the sample’s excitation (emission) dipoles [69]. This field-dependent PL is consistent with a spin-related inter-system crossing model, indicating optically addressable spin defects in hBN [69]. On the other hand, impurity defects in hBN have also been reported to exhibit bright single-photon emission [68,73,74,75]. Carbon impurity defects in hBN can emit single photons in the ultraviolet region [73].

Similarly, hBN can also be manipulated by applying strain to its emission wavelength [76]. As illustrated in Figure 3c, by applying localized strain through a flexible polycarbonate to hBN flakes, the emission wavelength of its SPEs can be reversibly tuned, with the maximum variation reaching up to 6 meV [56]. Notably, the sensitivity of different emitters to strain varies, primarily due to the differences in their initial strain (inset of Figure 3c). But, unlike TMD materials, some impurity defects in hBN have a large permanent out-of-plane dipole moment due to broken spatial inversion symmetry [68], allowing emission wavelength tuning via an applied out-of-plane electric field, with Stark shifts reaching 5.4 nm per GV/m [68]. Lastly, a novel method using ionic liquids for electrostatic tuning of emitters in hBN has achieved wavelength tuning up to 15 nm. The study reveals that using the ionic liquid BMIM-PF_6_ enables a rapid 17 nm wavelength shift without the need for gate voltage tuning, as depicted in Figure 3d [77].

## 3. Creation and Positioning of SPEs

Up to now, localized excitons or defect energy levels with single-photon emission properties in 2D materials have appeared randomly at the edges or folds of the material. This uncertainty in emission and the unpredictability of emission sites significantly impede the development of scalable quantum photonic devices. Consequently, various defect engineering and nano-etching techniques are now being employed to induce deterministic single-photon emission and precisely control its location in 2D materials. For example, annealing, chemical vapor deposition, and plasma etching can all generate SPEs on 2D materials, but the placement of these emitters remains uncontrollable. Thus, for precise manipulation, it is now common practice to use focused high-energy electron/ion beams produced by electron/ion microscopes [78,79], or lasers channeled through focused lenses, to irradiate samples, thereby accurately creating defects capable of single-photon emission. It is noteworthy that the defect engineering of many SPEs often requires a clean environment to prevent the deactivation of defects due to oxidation or the adsorption of other particles. Lastly, SPEs can also be deterministically generated on samples through strain engineering [40,80]. Additionally, one of the benefits of strain engineering is the ability to adjust the emission characteristics of the SPEs, while ensuring their deterministic production [81]. However, the key challenge in strain engineering is achieving a close and seamless integration of the substrate with the sample without damaging the latter [82,83].

### 3.1. Defect Engineering of SPEs

Initial investigations have demonstrated that annealing hBN materials in an inert gas setting or exposing them to electron irradiation under a low vacuum can engender localized defects endowed with the capacity for single-photon emission [67]. The proliferation of these defects correlates with escalating annealing temperatures, indicative of vacancies within their lattice [67]. Moreover, the stability of these emitters in a hydrogen atmosphere further corroborates the electrical neutrality of the defects [67]. Intriguingly, defects engineered through these two methodologies exhibit two distinct ZPL types. Yet, the PL spectra of these defects consistently show a PSB red-shifted by 160 ± 5 meV relative to the ZPL (Figure 4a), signifying that, despite the differences in ZPL types, the defects share a similar crystalline structure. Ion/atom implantation or irradiation has also been proven to effectively create optically active defects in hBN films, with post-irradiation samples exhibiting enhanced photostability [70,84,85,86]. For example, by oxygen irradiation, a localized emitter appears in hBN flakes with a sharp peek at 2.17 eV relative to the unirradiated sample, as shown in Figure 4b. Although the larger collision cross-section of ions may result in less precision compared to electron irradiation, a significant advantage is the ability to customize the type of defects by employing different ions [84]. Additionally, ion irradiation is applicable to TMD materials (Figure 4c); for instance, focused He ion irradiation on MoS_2_ can form high-density SPEs with axial and lateral precision of 0.7 nm/10 nm [55]. Encapsulating MoS_2_ with hBN before He ion irradiation can effectively reduce the influence of other adsorbates on luminescence [55]. Furthermore, plasma-etching and chemical-etching techniques are highly effective for inducing defects that facilitate single-photon emission [58,74,84], particularly in multilayer hBN samples. Slow-rate chemical etching is more conducive to forming SPEs compared to ion irradiation [84]. Additionally, employing oxygen plasma etching has successfully yielded SPEs with lifespans as brief as 294 ps [66], with their minimal longevity enabling a significantly higher operational bandwidth. Furthermore, short-pulse laser irradiation is an effective means of creating defects [70,87]; recent femtosecond laser irradiation of multilayer hBN has achieved quantum emitter arrays with up to 43% yield (Figure 4d) [87]. Compared to electron beam irradiation, defects formed by ion implantation, plasma etching, and ultrafast laser irradiation require subsequent annealing to be activated for single-photon emission.

### 3.2. Strain Engineering of SPEs

Applying localized strain on 2D materials can induce deformation potentials in specific areas, trapping free excitons and confining them at the points of maximum strain [83,88,89]. This results in the formation of three-dimensional, quantum dot-like confined excitons within the 2D system. Due to their isolation from the rest of the system, these three-dimensional confined excitons can facilitate single-photon emission.

This strain can be induced through various techniques, such as the protrusions of nanostructures [90,91,92], the gaps in nanorods [43,93], and the indentations on surfaces [57,94]. Interestingly, by employing techniques such as electron beam lithography to create periodically arranged nanopillar structures on a substrate, it is possible to achieve quantum array emission (Figure 5a). Meanwhile, variations in the strain gradient significantly impact the performance of SPEs [90,91]. For instance, increasing the height of the nanopillar can effectively reduce the spectral drift of quantum emitters. When the nanopillar reaches a height of 190 nm, the spectral drift among different quantum emitters in a WSe_2_ sample can be minimized to 0.1 meV (Figure 5b) [91]. Opting for a nanopillar with an aspect ratio of 0.3 maximizes the yield and precision of quantum emitters’ positioning [90]. Furthermore, in WSe_2_ materials, it has been observed that the polarization direction of quantum emitters tends to align with the direction of the strain field [95], providing an effective means for polarization control. For instance, by placing the material in the gaps between nanorods and utilizing the strain gradients induced by these gaps to generate SPEs, the polarization direction of emitted photons can be modulated by adjusting the size of the nanorod gaps (Figure 5c) [93]. This method of strain manipulation holds promise for addressing the randomness in the orientation of emission dipole moments in SPEs within 2D materials, thereby enhancing the coupling efficiency between the materials and optical devices.

Finally, it should be noted that bright excitons captured by strain-induced potential wells are not the sole source of SPEs in TMDs. Indeed, current theories posit that inherent defects in TMDs also play an indispensable role [49,50]. For instance, a study found that when a defect-free monolayer of WSe_2_ was transferred onto a nanopillar array, its photoluminescence spectrum did not exhibit the sharp emission peaks associated with single photons [50]. Therefore, the quantum emitters in some TMDs are likely the result of interactions between strain and atomic defects.

Surprisingly, this photoluminescence is closely associated with the dark exciton states. As shown in Figure 5d, the theory suggests that the deformation potential induced by strain tunes the energy of the material’s dark exciton states to levels close to its local defect states, resulting in the hybridization of exciton states. This disrupts the system’s valley selectivity, thereby enabling effective photoluminescence from the dark excitons. Recently, the luminescence of dark excitons induced by strain and defects has been observed in WSe_2_ through the use of nanoindentations created by an AFM probe [96].

## 4. The Property Enhancement of SPEs

Enhancing the interaction between light and matter has consistently been a focal point of research in fields such as photonics and quantum optics. Specifically, for unique systems like SPEs, enhancing their interaction efficiency with light, thereby increasing their brightness, represents a critical challenge. This is primarily due to the characteristics of their single-photon emission. Particularly for SPEs made from 2D materials, the intensity of light exhibits a distinct saturation effect as the excitation power increases [42,45]. Furthermore, under high-power excitation, the purity of certain SPEs significantly deteriorates [97]. Consequently, enhancing their brightness by increasing the excitation power is impractical. Fortunately, quantum electrodynamics informs us that the radiative properties of an emitter can be modulated by altering the surrounding electromagnetic field environment. This is the essence of the Purcell effect: when an emitter is placed within a microcavity and its emission frequency matches the resonance frequency of the microcavity, the emitter’s spontaneous emission rate is enhanced. This increase is primarily due to the reduction in the mode volume of the electromagnetic field within the microcavity, leading to an increase in the local density of photonic states. The degree to which the Purcell effect enhances spontaneous emission is quantified by the Purcell factor (F_P_), defined as the ratio of the spontaneous emission rate within the microcavity (γ_c_) to that in free space (γ_sp_). Typically, the Purcell factor is directly proportional to the quality factor (Q) of the microcavity and inversely proportional to the mode volume of the cavity. Finally, for 2D materials, the emission characteristics of the dipole in their SPEs, combined with the ultra-thin layer thickness inherent in 2D systems, make them particularly well-suited for coupling with various types of cavities compared to other systems. SPEs of 2D materials have been successfully coupled with various optical devices, including plasmonic cavities and developed optical microcavities [33,98].

### 4.1. Plasmonic Field Coupling with SPEs

In plasmonic cavities, plasmons in conductors and dielectrics can concentrate the optical field to sub-wavelength scales [99]. Typically, the field of surface plasmons decays within tens of nanometers from the conductor’s surface, necessitating precise alignment between the localized emitters and the plasmonic cavity. A common approach is to deposit 2D materials directly onto silver nanowires or plasmonic nanopillars (Figure 6a) [99,100,101,102]. Here, the silver nanowires or gold–silver nanoparticle arrays not only provide localized strain to excite single-photon emission but also enable the excited single photons to couple directly into the surface plasmon field, eliminating the need for alignment steps. Studies have shown that the saturation intensity of hBN localized emitters placed on silver nanoparticle arrays increased by 2.6 times (Figure 6b) [99], with an overall Purcell factor of about 2 [99]. However, these nanowire and array structures face a challenge due to the strong perpendicular electric field component of surface plasmons relative to the metal surface [101], while the SPEs in 2D materials exhibit in-planar dipole emission characteristics, with their electric field along the tangential direction of the metal surface. This misalignment between the electric field of the localized emitters and the surface plasmons leads to low coupling efficiency. Recently, metal–insulator–metal waveguides have been effective in resolving this misalignment issue [31]. Additionally, since the emission wavelengths of localized emitters in 2D materials typically have a broad distribution, enhancing the quantum emitter array through the Purcell effect requires the plasmonic cavity’s Q factor to be not too high [99], providing a wider resonance peak range to match the different emission wavelengths of various emitters, which significantly limits the enhancement effect.

### 4.2. Optical Microcavity Coupling with SPEs

To facilitate Purcell enhancement, an alternative strategy involves coupling the light field of an SPE with the mode field of an optical microcavity [103,104,105,106,107]. Composed of a hemispherical mirror and a flat mirror (Figure 6c), the quality factor of such a microcavity can vary from hundreds to thousands by precisely controlling the radius of the hemispherical mirror. Studies have demonstrated that placing a monolayer SPE of WSe_2_ within an optical microcavity can increase the spontaneous emission rate by 7.8 times, with the saturation intensity rising from 74 kcts/s to 332 kcts/s [105]. One advantage of optical microcavities is the ability to adjust the cavity length, altering the resonance peak frequency to match emitters of different wavelengths [32,105]. Generally, the Q factor of optical microcavities used for Purcell enhancement is very high, leading to sharp resonance peaks that require perfect coupling with the emission wavelength of the quantum emitter [32,105,108]. Although the frequency of the resonance peak can be flexibly adjusted, it is still not suitable for integrating large-scale quantum emitter arrays. Recently, circular Bragg grating bullseye cavities (CBGBs) have shown promise in overcoming this challenge, as shown in Figure 6d; when the SPE in WSe_2_ is coupled to the CBGBs, its spontaneous emission lifetime is reduced to 0.6 ns [108].

**Figure 6 nanomaterials-14-00918-f006:**
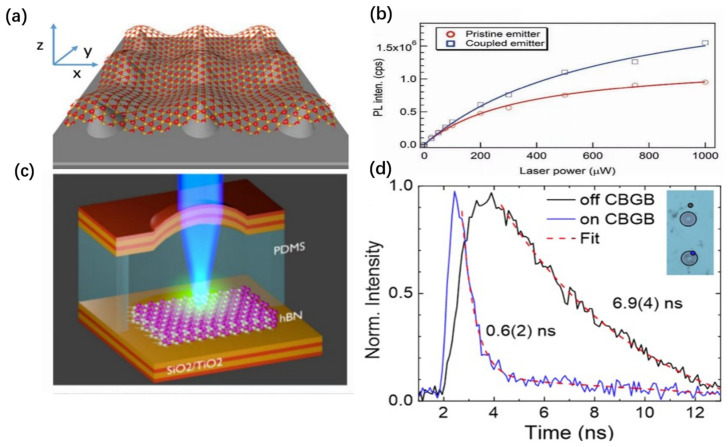
Purcell effect of SPEs in 2D materials. (**a**) Artistic sketch of the assumed sample configuration, featuring the monolayer wrapped over metal cones evolving on the silver–Al_2_O_3_ surface. (**b**) Fluorescence saturation curves obtained from the pristine (red open circles) and coupled (blue open squares) systems. (**c**) The microcavity consists of a hemispherical and flat mirror (only two stacks shown on either side). The quantum emitter hosted by hBN emits confocally with the excitation laser. A PDMS spacer sets the cavity length. To prevent the polymer from influencing the emitter, the PDMS is etched in the middle. (**d**) Normalized lifetime measurement on (blue) and off (gray) the CBGB. The values 0.6(2) and 6.9(4) ns indicate the extracted lifetime of the transitions, respectively. Panel (**a**) reprinted with permission from Ref. [102], Copyright 2018, American Chemical Society. Panel (**b**) reprinted with permission from Ref. [100], Copyright 2017, American Chemical Society. Panel (**c**) reprinted with permission from Ref. [32], Copyright 2019, American Chemical Society. Panel (**d**) reprinted with permission from Ref. [109], Copyright 2021, American Chemical Society.

## 5. Conclusions and Outlook

Due to their exceptional performance, SPEs in 2D materials are increasingly recognized as promising quantum light sources, with the potential to broaden the applications of 2D materials. Concurrently, the field of 2D materials is witnessing an acceleration in the research of SPEs, propelled by advanced nanomanipulation techniques and characterization methods. However, current research predominantly focuses on a select few 2D materials, and the properties of certain defects within these materials remain inadequately explored. For instance, the underlying causes of biexciton cascade emissions, as reported in some studies, are still not well understood. For TMDs, their narrow bandgap results in the defect exciton energy levels being close to the electron energy levels. Consequently, at high temperatures, the single-photon emission properties dissipate, necessitating the elevation of the operational temperature of SPEs or the discovery of alternative materials that exhibit a stable single-photon emission at room temperature. Similar to hBN, other group III nitrides are also considered ideal for fabricating SPEs [109]. Although most single-photon emissions have been observed in bulk materials, the anti-bunching effect has recently been discovered in monolayer InGaN islands [110]. Moreover, defects in monolayer h-AlN have been theoretically demonstrated to possess single-photon emission capabilities [16]. Additionally, the rich excitonic properties inherent in the recently emerged moiré superlattices make them well-suited for single-photon studies [111,112]. Moreover, moiré excitons have already demonstrated characteristics of single-photon emission. Although they operate at low temperatures, some studies have shown that by modifying factors such as the twist angle to deepen the moiré potential, the dissociation energy of excitons can be effectively enhanced [113]. This provides a novel research direction for achieving single-photon emission at room temperature. However, the challenging aspect lies in the precise control of the twist angle in moiré superlattices, making it exceedingly difficult [114], particularly given the current lack of effective methods for the in situ monitoring of these structures.

Additionally, SPEs in 2D materials are confronted with the issue of insufficient purity. For quantum key distribution, which demands the lowest purity, it is required that g^2^(0) ≤ 0.1. Unfortunately, the current SPEs in 2D materials typically exhibit a g^2^(0) ranging from 0.2 to 0.4. The primary sources of low purity are the background luminescence caused by additional defects or other impurities, which spectrally overlap with the luminescence of quantum emitters, making it challenging to filter out via spectral filtering. Furthermore, for hBN, the influence of phonon sidebands results in an asymmetric line shape of its zero-phonon line, leading to a certain broadening and a significant background presence. Numerous methods have been developed to suppress these background noises, such as effectively reducing the background luminescence of additional adsorbates produced after ion bombardment by encapsulating MoS_2_ with hBN. Additionally, optical microcavities also can effectively suppress phonon sidebands and other non-resonant noise in hBN, thereby enhancing the purity of single photons [32]. Experiments have shown that the g^2^(0) of the coupled hBN single-photon source decreased from 0.051 to 0.018 [32]. Indeed, the HOM experiments involving SPEs in 2D materials merit attention, as they are crucial for assessing the indistinguishability of photons, which is a key parameter for quantum computing and communication applications. Yet, there is a paucity of literature on HOM interference experiments with SPEs in 2D materials, primarily because the emission wavelengths of most SPEs in such materials are subject to spectral diffusion, blinking, or quenching, which significantly reduces the indistinguishability of the photons. The fluctuations in the emission wavelength primarily stem from the instability of the environment surrounding the emitter. For instance, when inducing single-photon emission through a nanopillar, the inability of the material to conform tightly to the nanopillar leads to fluctuations in local strain, thereby affecting the emitter’s stability. A contemporary approach that synergizes ‘top-down’ and ‘bottom-up’ methodologies offer a controllable wrinkling technique to effectively address issues of poor adhesion. Finally, the random distribution of emission wavelengths in SPEs within 2D materials is a significant barrier to producing indistinguishable photons. The current primary tuning methods are unable to encompass their extensive distribution range. Moreover, for TMDs, the strong in-plane dipole moment of their excitons precludes effective wavelength modulation through external electric fields. Fortunately, for 2D materials, the interlayer excitons within their moiré superlattices appear capable of achieving this [61,62].

Over the past decade, 2D materials have revolutionized the development of SPEs, particularly due to their dangling-bond-free structural characteristics that allow scientists to stack different materials like Lego bricks, creating moiré superlattices with unique physical properties. These superlattices exhibit a rich array of optical characteristics, offering new possibilities for single-photon technologies. However, current research remains primarily in the experimental phase, with the potential of moiré photonics and optoelectronics far from being fully realized. Future breakthroughs in this field are anticipated to be highly promising.

## Figures and Tables

**Figure 1 nanomaterials-14-00918-f001:**
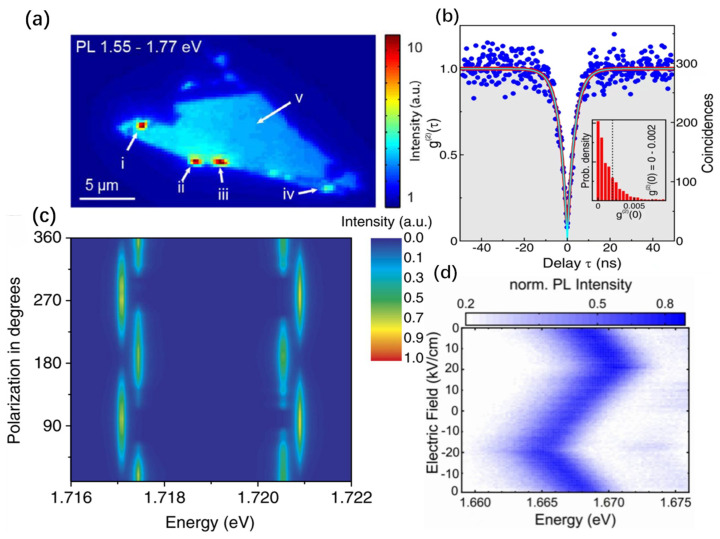
SPEs of WSe_2_. (**a**) PL intensity map of the broad spectral range 1.55–1.77 eV, (i) (ii) (iii) (iv) (v) for localized exciton luminescence appearing randomly on a monolayer WSe2 sample. (**b**) g^2^(τ) under resonant CW excitation of the BS-*X*. (**c**) Contour representation of the four peaks, after normalizing to the maximum peak intensity, yielding a fine structure splitting ~0.4 meV. (**d**) Contour plot of the µPL spectra of an SPE as a function of the applied electric field on the piezoelectric actuator. The electric field is reversibly swept in the range from −20 to 20 kV/cm. The observed red- and blue-shifts are due to the induced compressive and tensile strain fields by the actuator. An energy shift equal to 5.4 μeV/V is observed. Panel (**a**) reprinted from Ref. [42], Copyright 2015, Optical Society of America. Panel (**b**) reprinted from Ref. [51], Copyright 2016, Optical Society of America. Panel (**c**) reprinted from Ref. [52], Copyright 2016, the author(s). published by Springer Nature. Panel (**d**) reprinted with permission from Ref. [53], Copyright 2019, American Chemical Society.

**Figure 3 nanomaterials-14-00918-f003:**
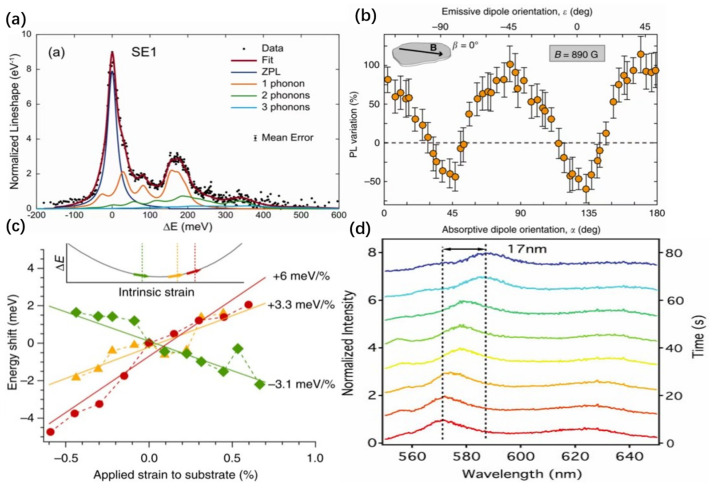
SPEs of hBN. (**a**) Emission line shape for SE1 in hBN, as a function of the change in lattice energy during optical relaxation (ΔE = E_ZPL_ − E, where E is the observed photon energy). The data are binned to produce approximately uniform uncertainty, as indicated by the representative error bar. Curves show the results of a fit using the model described in the text (thick red curve), along with the ZPL and PSB components (thin curves) as indicated by the legend. (**b**) PL variation as a function of the relative orientation between the emitter’s optical dipoles and an in-plane B = 890 G. (**c**) The plot shows the scaled energy shift as a function of applied strain to the bendable substrate for three emitters with different tunability values of −3.1 meV/% (green), +3.3 meV/% (yellow), and +6 meV/% (red). Inset shows a sketch of a quadratic energy shift, ΔE, for the single-photon emission induced by intrinsic strain. (**d**) A gradual 17 nm red shift is observed by 200 μW, 532 nm CW illumination to the sample; there is no gate voltage applied to induce this effect. The different colored lines correspond to the PL spectral lines of the samples measured at different times. In this case, the electrons are liberated from the ionic liquid by the laser excitation alone using a λ_exc_ = 532 nm illumination source at 200 μW. The spectra are collected for 10 s each during a time-resolved PL measurement and offset for clarity. The time axis maps the collection time of the spectra displayed. Panel (**a**) reprinted with permission from Ref. [65], Copyright 2017, American Chemical Society. Panel (**b**) reprinted from Ref. [69], Copyright 2019, the author(s), published by Springer Nature. Panel (**c**) reprinted from Ref. [76], Copyright 2017, the author(s), published by Springer Nature. Panel (**d**) reprinted with permission from Ref. [77], Copyright 2019, American Chemical Society.

**Figure 4 nanomaterials-14-00918-f004:**
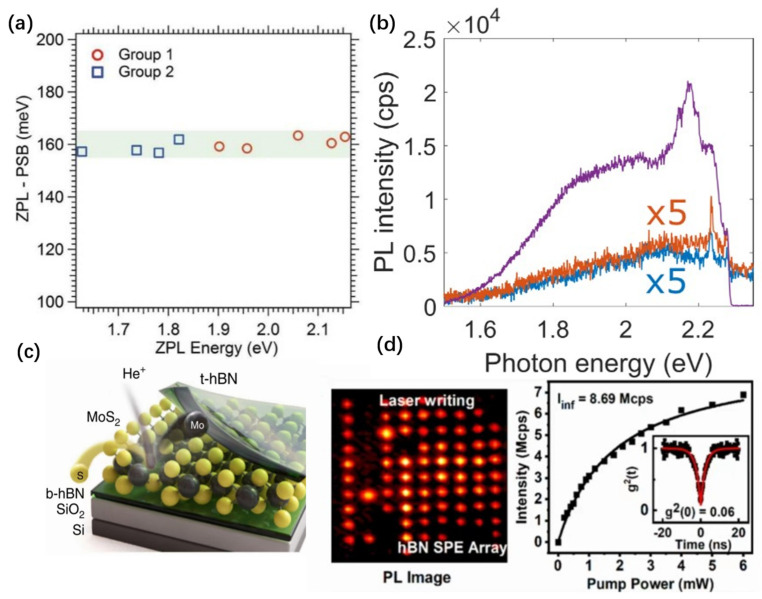
Defect engineer of SPEs. (**a**) Difference in the energy of the ZPL and PSB versus ZPL energy. (**b**) Spectral evolution of an individual luminescent center. The purple line shows the PL spectrum after oxygen irradiation and annealing, the blue line represents the untreated sample, and the orange line indicates the PL spectrum after only oxygen irradiation. The spectra after exfoliation and pure irradiation are multiplied by 5. All data shown are obtained from hBN flakes of batch 1 irradiated for 10 s at 240 eV. (**c**) Schematic illustration of the exposed MoS_2_/hBN van der Waals heterostructure. (**d**) The left image shows the PL image of SPEs in hBN sample following irradiation by a femtosecond laser pulse. The right image details the variation in light intensity of an individual SPE as a function of the excitation power, with an excitation wavelength of 514 nm and a beam size of 1 μm. The black line represents the fitted results, The inset graph depicts the g^2^(τ) curve obtained from an HBT experiment for a single emitter. Panel (**a**) reprinted with permission from Ref. [67], Copyright 2016, American Chemical Society. Panel (**b**) reprinted with permission of AAAS from Ref. [86]. Copyright 2021, the authors, some rights reserved, exclusive licensee AAAS. Distributed under a Creative Commons Attribution Noncommercial License 4.0 (CC BY-NC). Panel (**c**) reprinted from Ref. [58], Copyright 2019, the author(s), published by Springer Nature. Panel (**d**) reprinted with permission from Ref. [87], Copyright 2022, American Chemical Society.

**Figure 5 nanomaterials-14-00918-f005:**
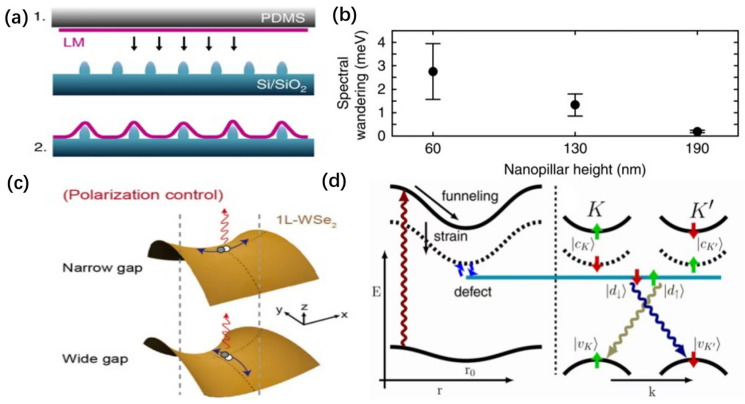
Strain engineer of SPEs. (**a**) Illustration of the fabrication method: (1) mechanical exfoliation of LM on PDMS and all-dry viscoelastic deposition on patterned substrate; and (2) deposited LM on patterned substrate. (**b**) Increasing nanopillar height also leads to a reduction in spectral wandering. Solid black circles represent the mean value of spectral wandering of several QEs for a given nanopillar height, while the error bars represent the standard deviation of each distribution, both extracted from time-resolved high-resolution spectral measurements. (**c**) Schematics of the deformed monolayer WSe_2_ due to the nanogap. The saddle-shaped deformation occurs along the x-axis (y-axis) for the narrow(wide) nanogap. The exciton oscillation is aligned with the elongation direction. (**d**) Real space representation: A free exciton is created (dark red arrow), and the strain efficiently funnels excitons with the electron in the bright (solid black line) and dark (dashed black line) conduction band down in energy towards the strain maximum near r_0_ due to the strain-dependent band gap. The red and green arrows indicate the different spin directions of the electrons. Mixing of the strain-localized dark exciton with a defect state leads to the formation of a strongly localized defect exciton. Panels (**a**,**b**) reprinted from Ref. [91], Copyright 2017, the author(s), published by Springer Nature. Panel (**c**) reprinted with permission from Ref. [93], Copyright 2021, American Chemical Society. Panel (**d**) right adapted with permission from Ref. [49], Copyright 2019, APS.

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
