# Peer review of "Research Progress of Single-Photon Emitters Based on Two-Dimensional Materials"

_nanomaterials, 2024, doi:10.3390/nano14110918_

Round 1

Reviewer 1 Report

Comments and Suggestions for Authors

The work is very informative, and clearly written with good English, I found only a couple f issues to quibble with,

1. What looks like a repeated line

By capitalizing on 37

the information encapsulated within intrinsic quantum systems, an array of quantum 38

technologies with vast potential is burgeoning, encompassing quantum informatics, 39

quantum metrology, and quantum imaging, among others [1-4]. A burgeoning suite of 40

quantum technologies is emerging, spanning quantum informatics, quantum metrology, 41

and quantum imaging

2. A double plural

Interestingly, in TMDs materials- 122

maybe better as either TMD materials (as used in line 132)

or just TMDs

Author Response

Thank you very much for your comments, we have responded to each comment individually and the results are in the attachment

Reviewer 2 Report

Comments and Suggestions for Authors

The manuscript presents an exceptionally well-written, timely, and comprehensive review on quantum emitters based on 2D materials. The papers is very well organised covering all the important aspects of 2D materials single photon emitters, while includes a thorough critical discussion on their quantum performance further to simply material aspects.

I have no comments that could significantly improve the quality of the paper, and I recommend the publication of the paper.

However, I would suggest to authors to provisionally consider the inclusion in the Introduction of a section or paragraph where it could be discussed the performance, the comparative assessment, and the anticipated penetration in quantum applications of major alternative quantum emitters such as quantum dots, semiconductor nanowires with quantum dots etc. That would help the readers to form a generic view and the perspectives of quantum emitters’ development in relevant applications

Author Response

(The authors gave the same response as above.)
